# Using Indicators to Evaluate Cultural Heritage and the Quality of Life in Small and Medium-Sized Towns: The Study of 10 Towns from the Polish-German Borderland

**Sławomir Książek** [1], **Magdalena Belof** [2,*], **Wojciech Maleszka** [1], **Karolina Gmur** [1], **Marta Kukuła** [1], **Robert Knippschild** [3,4], **Eva Battis-Schinker** [4], **Bettina Knoop** [3] and **Sarah Al-Alawi** [3,4]

1   Institute for Territorial Development, Dawida 1a, 50-527 Wroclaw, Poland; slawomir.ksiazek@irt.wroc.pl (S.K.); wojciech.maleszka@irt.wroc.pl (W.M.); karolina.gmur@irt.wroc.pl (K.G.); marta.kukula@irt.wroc.pl (M.K.)
2   Faculty of Architecture, Wroclaw University of Science and Technology, Wybrzeże Stanisława Wyspiańskiego 27, 50-370 Wroclaw, Poland
3   International Institute (IHI) Zittau, Technische Universität Dresden, Markt 23, 02763 Zittau, Germany; r.knippschild@ioer.de (R.K.); bettina.knoop@tu-dresden.de (B.K.); al-alawi@outlook.de (S.A.-A.)
4   Leibniz-Institute of Ecological Urban and Regional Development (IOER), Weberplatz 1, 01217 Dresden, Germany; evabattis@gmail.com
*   Correspondence: magdalena.belof@pwr.edu.pl; Tel.: +48-606-997-613

**Abstract:** Cultural built heritage (CBH) is one of the most important cultural assets that affect the quality of life (QoL) in cities, and it is especially significant in small and medium-sized towns that lack some other advantages of larger urban centres. However, in quality of life studies, CBH is often neglected or treated superficially. This is probably due to the lack of a universal method developed to study their mutual interdependence based on a set of proven, objective indicators. This paper presents the authors' attempt to fill this methodological gap by developing a set of indicators that would make it possible to assess the relationship between QoL and CBH. The study focused on historic small and medium-sized towns, located peripherally on the borderlands between Poland and Germany, since it was considered that especially for such towns, the architectural and urban heritage can appear as a crucial factor in improving the quality of life. To develop a comprehensive understanding of the mutual relationship between the phenomenon of QoL and CBH, a triangulation of research methods has been adopted: first, a literature review, where indicators were sought; second, semi-structured interviews and workshops with selected experts; and third, focus-group studies in 10 pilot locations. The analysis yielded quantitative and qualitative indicators in each set, allowing for the measurement of the mutual relationship between QoL and CBH. The set offered a total of 20 indicators developed specifically for small and medium-sized towns, and despite certain limitations, it can be considered universal and can also be applied to other urban centres. Furthermore, the study identified the five distinct spheres in which the relationship between QoL and CBH can be observed and analysed. This typology can be used as a background for deeper studies at specific sites, regardless of their size and location.

**Keywords:** cultural built heritage; quality of life; SMTs; indicators; Lower Silesia; Saxony

## 1. Introduction

The socio-economic processes of recent decades, in particular the rise of metropolises, large-scale migration, an ageing society, and economic restructuring, may pose a threat to the development of small and medium-sized towns (SMTs) [1]. These trends make such settlements unable to compete on many levels with large urban centres, which are at the heart of development processes. The growing imbalance and polarisation in development between the largest cities and the peripheral areas have been termed the "backwash effect" [2]. This places pressure on SMTs, especially those located in the periphery of the largest national centres. The situation is even more critical in countries or regions

previously within the Soviet Bloc, such as Poland or East Germany, where global trends are exacerbated by the post-socialist processes of economic and political transformation. Yet, despite the observed concentration of many negative factors such as ageing populations, few attractive and well-paying jobs, and the dearth of culture and entertainment, smaller urban centres have undoubted advantages that positively affect their QoL. These include a lower cost of living, a more peaceful urban environment, a stronger sense of community, and the ability of residents to directly influence local affairs. Among those important factors that contribute to a better quality of life and the socio-economic situation in SMTs, one has been mostly overlooked in research: the cultural built heritage (CBH). In particular, this seems to play a vital role in smaller towns with a rich historic urban structure because, to a certain extent, its presence can compensate for a lack of other assets found in larger cities.

The border area between Lower Silesia (Poland) and Saxony (Germany) has a very dense, historically formed settlement network characterized by the existence of many small and medium-sized towns of medieval origin, with remarkable urban structures and a significant architectural heritage. Most of them face developmental problems due to their peripheral location in relation to large regional and national centres (Wroclaw, Dresden, Berlin). In both countries, Poland and Germany, decision makers at the national, regional, and local levels (e.g., [3–5]) are aware of the negative social and economic trends which affect SMTs on the periphery, as well as of the urgent need for a long-term development strategy to maintain a polycentric settlement structure, which has developed throughout the centuries [2,6,7]. The construction of these strategies very rarely emphasizes the importance of the so-called soft factors of QoL, including CBH, which, if properly used and managed, can significantly influence development trajectories and affect QoL in these centres. This lack of political interest in the reciprocal relationship between QoL and CBH is due, among other things, to the fact that there is no research directly indicating a link between the two spheres, and to date, no universal set of indicators has been developed to measure them.

The main objective of carrying out this study was to fill the research gap and contribute to a methodology that would allow, in a more appropriate way, the assessment of QoL in historical SMTs located in peripheral locations, taking into particular account the perspective of their cultural built heritage. The study was based on the assumption that the way in which urban QoL is currently measured does not sufficiently take into account specific soft location factors of historic SMT with a pronounced CBH. Another assumption was that the urban and architectural legacy and the associated immaterial heritage of historic towns make a significant contribution to urban quality of life, but so far, this has received too little attention in QoL research.

## 2. CBH as a Parameter of Quality of Life in Small and Medium-Sized Towns

Quality of life is a term used in various academic fields such as psychology, sociology, medicine, philosophy, and geography. The development of modern research on QoL started in the 1970s and is generally associated with the work of Angus Campbell. Due to the various notions of QoL applied in different disciplines and contexts, there is currently no unambiguous, commonly accepted definition [8]. Moreover, terminological confusion is compounded by the interchangeable use of the concept of quality of life with other terms such as standard of living, living conditions, or well-being [9]. Some authors prefer to use the term "standard of living" in relation to objective QoL: thus, a clear distinction is made between the standard of living determined by objective measures and the QoL based on subjective indicators. In fact, these are treated as two separate categories [10].

Quality of life can be assessed subjectively or objectively [9,11,12]. Wider studies, whether regional or national, along with international comparisons, are usually based on subjective or objective indicators, depending on their particular focus. The majority of the studies tend to rely on comparable statistical data such as the GDP, unemployment rates, and population education level, as well as the evaluations of free time, education, civic participation, or living conditions; however, some value other parameters that are more suited to the studied concept (i.e., the Human Development Index). Even in the

most extensive approaches to measuring quality of life, such as the one proposed by the OECD [13], the built environment is represented only through a few selected and measurable aspects [13] that are disjointed from the cultural heritage perspective. In some studies related to life satisfaction, built and social capital are even jointly assessed [14].

A subjective approach is generally observed in smaller studies focusing on a particular area or individual town [15–21]. These usually adopt more diverse sets of indicators in accordance with a thematic orientation (e.g., QoL in small cities; see study [20,21]). These studies more often stress aspects related to the landscape and built environment in the city using adequate indicators; however, CBH still does not seem to play a particular role in the assessment of the quality of life. This can be attributed to the difficulties in transforming theoretical considerations into practical measures.

The concept of sustainable development introduced a new perspective on the assessment of the quality of life [22,23]. In recent years, it has received rich examination in the scientific literature, and numerous indicator approaches have been developed (i.e., [24,25]). Research on sustainable development takes into account the aspects of the built environment, landscape, and cultural heritage in a clearer and more direct way than research on the quality of life [26]. It was recognized that cultural heritage as a common good can "produce benefits for the community in which it places. Therefore, tools for assessing the impacts of cultural heritage conservation/valorization on community wellbeing are necessary" ([27], p. 22).

Scholars proposed various insights and indicator-based approaches to better understand the mutual relationship of sustainable development and CBH. Tweed and Sutherland [25] propose three groups of indicators to assess the perceptions and attitudes of citizens to the built heritage in towns and cities; Berthold, Rajaonson, and Tanguay [28] proposed an indicator-based approach to capture the interconnection between sustainable development and urban heritage management. Their work aims at a better understanding of the dynamics between development and conservation, two domains that for a long time were regarded as opposing practices. Nocca [27], who identified 177 indicators linking cultural heritage with sustainable development, claims that "the contribution of cultural heritage to wellbeing is poorly considered", since "only seven indicators out of 177 emerge in this category" ([27], p. 23).

Appendino ([29], p. 10), who conducted a review of studies focusing on heritage indicators for sustainable urban development, discovered a substantial gap between cultural heritage management and sustainable development, observing that "although culture-related indicators are gaining ground within urban sustainability assessment tools and methodologies [ . . . ] in practice, there is still a limited number of indicators and that heritage is not yet considered in all its complexity and potential".

Guzman et al. ([30], p. 4) also stress that the identification and use of common indicators "can contribute to the understanding of those urban synergies between the conservation of heritage as a cultural resource and wider factors for development that aim to thrive whilst achieving sustainability" The recent strong scientific focus on sustainable development and its measurements increases the interest in CBH; however, it does not offer any answer on how cultural heritage influences QoL in towns. Nevertheless, this contribution seems to be indisputable. According to a study by the European Commission published in 2017, more than seven out of ten Europeans believe that cultural heritage can improve their quality of life [31]. This impact can be analyzed along various dimensions (as will be discussed in more detail in Section 4).

Cultural built heritage can be particularly important to smaller, peripheral urban centres, where many other services and economic benefits are either lacking or much weaker than in larger cities. This potential seems to be confirmed by Guzman et al. [30] who observe that "in the last twenty years, the cultural heritage's role in urban management has evolved from institutionalizing conservation efforts to placing heritage at the focus of strategic planning" ([30], p. 1).

However, review studies on urban QoL typically do not differentiate between the size of centres, thus failing to take into account the very different living conditions between

small and medium-sized towns and large metropolitan areas. One difficulty facing research specific to SMTs is the lack of any agreed definition of "small" and "medium-sized" towns [32–34]. In particular, the size range for medium-sized towns is found to vary considerably; this may also explain the relatively low research interest in such towns compared to small settlements and large cities [35]. In both Poland and Germany, the most commonly used measure of town size is the number of inhabitants. The Polish authorities [36] and Germany's Bundesinstitut für Bau-, Stadt-, und Raumforschung [37] divides urban settlements into three categories:

- — small towns of up to 20,000 residents;
- — medium-sized cities of 20,000–100,000 residents;
- — large cities of at least 100,000 residents.

At the beginning of the 21st century, a great deal of research was conducted in Poland and Germany on the process of metropolisation, metropolitan functions, and the formation of metropolitan centres and regions along with their networks. However, more recently, we can observe an increasing number of publications focused on small and medium-sized towns [6,7,32,35,38–51]. Moreover, this trend seems to be international in scope [33,34,52–56], undoubtedly driven by the search for new roles for smaller urban centres in a globalised world after they have lost their centuries-old traditional functions.

Well-developed and properly functioning SMTs are vital elements of a polycentric settlement structure [2,7,34,50,57,58]. Under the impact of structural change, many SMTs are now being forced to seek new identities [59]. Of course, these urban settlements do not form a homogeneous class: some are well-developed centres that have retained their traditional functions and may even have gained new ones. Others, mostly located on the periphery or characterised by long-term monofunctional development, have lost some previously dominant functions under socio-economic pressures. Such SMTs suffer from a number of social, economic, and spatial problems. The role of SMTs (especially small towns) as intermediaries of various socio-economic activities and processes is decreasing [60]. These services can now be transferred from the largest centres directly to rural areas, thanks to information and communication technologies. SMTs are particularly affected by negative demographic trends such as ageing populations and the outflow of younger and educated residents to larger cities. These processes serve to weaken local labour markets, municipal budgets, and social infrastructure.

The literature review allowed two main observations: first, the existing scientific literature and specialized studies do not adequately reflect the specific qualities of CBH as an element contributing to the quality of life. As a consequence, no adequate set of indicators is offered in this regard. Second, there is no specific research on the QoL phenomenon in small and medium-sized towns that would differentiate between and stress alternative components to the QoL in large cities. Therefore, the unification of indicators for the general "urban" quality of life can decrease the value of CBH.

These observations point to a clear research gap surrounding this issue, which, to some extent, this study intends to reduce.

Conceptually, this study is based on the following assumptions:

1. Multifaceted relations between CBH and quality of life exist and can be identified, structured, and measured by indicators. At the same time, the various relations between these phenomena are often intuitively understood and indirectly discussed in numerous studies rooted in various scientific disciplines.
2. The relations between CBH and QoL are more visible in small and medium-sized towns lacking other assets characteristic of larger cities.

The combination of the two above assumptions led us to the elaboration of the methodological structure explained in the following section.

### 3. Research Methods

This section briefly sets out the methodological framework we applied in our research process in order to best approach the complex relations between QoL and the CBH. It also introduces the 10 towns selected for pilot studies.

To improve the understanding of the QoL phenomenon and the possibilities of its measurement and to develop a comprehensive set of adequate indicators, the triangulation of research methods was used. This approach enabled us to confront and cross-analyse the findings from different research methods and data sources. Our research methods included:

1.  The literature review, which focused on three thematic areas: QoL, SMTs, and CBH. The relationship between these fields was analysed with the aim of identifying gaps in the current quality of life assessments of historic small and medium-sized towns. From the large number of literature sources, we extracted 22 studies focusing on the quality of life [16–21,61–76], (see Table A1 in Appendix A). Since the empirical part of our research was aimed at towns located in the transborder area, we focused in particular on studies that are nationally recognized as important for Polish and German urban policy and research. The studies were found, with few exceptions, using internet search engines; no results have been excluded, as long as we considered them to be a studies on the quality of life in cities (some additionally focus on the region). The studies were first selected separately by the German and Polish members of the team and then jointly discussed and approved. The final set of analyses included 11 studies from Germany, 7 from Poland, 1 from Italy, and 3 with an international focus. The Polish and German studies, which were of primary interest to the authors, come from the years 2006–2018. The purpose of the analysis was to explore the extent to which the characteristics relevant to the QoL of historic SMT (in peripheral locations) are taken into account in the evaluation of urban QoL, particularly in Germany and Poland. Most of these studies focused on urban areas, but several also considered the entire country or region without differentiating between urban or rural areas. From 22 studies on QoL, we identified and extracted 412 indicators that we assigned to 10 categories (dimensions) of QoL, i.e., work, living conditions (housing), services for citizens, free time, nature and cultural heritage, health, safety, education and engagement, and society. In the next step, the ten dimensions were compressed into five that better relate to the object of study without limiting the overall scope of the issue. (Table 1).

2.  Semi-structured interviews and expert workshops. For the scope of the study, we conducted an extended interview (in written form) with five selected German and Polish experts representing the fields of urban regeneration, SMT research, QoL, innovative indicators, and built heritage. In the first phase, the role of the experts was to indicate and assess possible relations between QoL and CBH, as well as to propose the indicators for their measurement according to their fields of expertise. The experts' opinions, together with the findings of the authors' literature review, were then confronted and jointly discussed during the workshop. All the results were then summarised, and on this basis, the assumptions on a possible contribution of the built heritage to quality of life in small and medium-sized towns (one attributed to each of the five dimensions mentioned above) were formulated. The third task of the experts was to formulate their opinions on the proposed assumptions (the final set of dimensions and assumptions attributed to them are presented in Table 1).

3.  The focus group discussion served as the platform for confirming or falsifying the assumptions elaborated previously. The assumptions were examined in 10 small and medium-sized towns on the Polish -German borderland. The pilot towns were the partner towns of the Polish-German project "REVIVAL!—Revitalization of Historic Towns in Lower Silesia and Saxony" implemented under the Interreg Poland-Saxony 2014–2020 cooperation programme: Bolesławiec, Chełmsko Śląskie (Chełmsko Śląskie lost its status as a town in 1945, after which it has been treated administratively

as a village. However, because the settlement has retained its historic town centre with numerous elements of cultural heritage, the authors decided to include it in the study. In this article, the term "town" is used to refer to all 10 settlements, including Chełmsko Śląskie), Gryfów Śląski, Kamienna Góra, Lubomierz, and Żary in Poland; and Bautzen, Görlitz, Reichen-bach/O.L., and Zittau in Germany (Figure 1). All pilot towns possess a historic centre that is structurally distinct from other areas in terms of urban functions and fabric, in particular, an abundant CBH. All 10 towns are of medieval origin, having obtained their town charters in the 13th or 14th centuries. Of the six Polish towns, Żary belongs to the Lubusz Voivodeship, while the rest are located in the Lower Silesian Voivodeship. All four German towns are located in Saxony. According to data from 2019, the towns that could be classified as medium are: Görlitz (55,980 inhabitants), Bolesławiec (38,872), Bautzen (38,425), Żary (37,304), and Zittau (25,085). Only the small town of Kamienna Góra (18,840) has a population in the range of 10,000 to 20,000. The remaining towns have less than 10,000 inhabitants, including Gryfów Śląski (6617), Reichenbach/O.L. (4915), Chełmsko Śląskie (1936 (Data for Chełmsko Ślaskie from the PESEL registry)), and Lubomierz (2004) [77,78]. There is also great diversity in the area size of the settlements, ranging from 8 km$^2$ to 67 km$^2$, as well as in the size of their historic centres. From a regional and national perspective, most of the towns can be described as peripheral. Furthermore, they are also largely peripheral in socioeconomic terms. The meetings were held between October 2019 and January 2020 and were attended by representatives of the town administrations and societies of the partner towns. The composition varied and at times included mayors, elected town representatives, associations, traders, educational institutions, heritage conservation agencies, town managers, and planners, all of various age groups. The discussion focused on the links that exist between cultural heritage and local quality of life in these towns. Each of the five proposed dimensions was discussed along with its associated assumption. The selected results of the focus group discussions, along with literature references, are presented in Section 4.

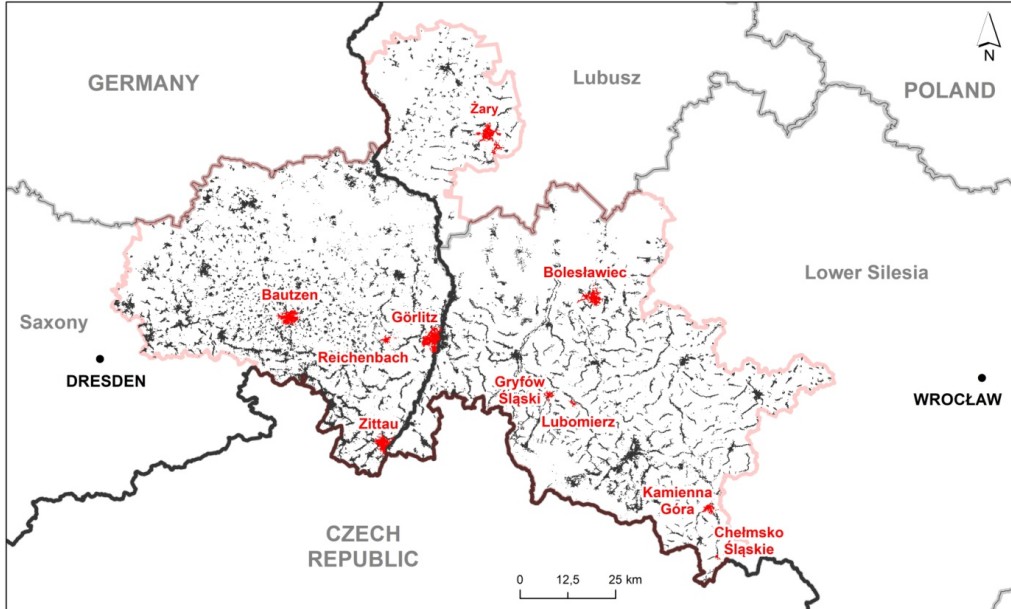

**Figure 1.** The partner towns of the REVIVAL! project in the area supported by the cooperation programme Interreg Poland-Saxony, 2014–2020. Source: Our own elaboration based on the use of OpenStreetMap.

**Table 1.** The dimensions of interdependence between cultural built heritage and quality of life.

| Dimension | Assumption |
|---|---|
| 1. Urban Identity/Sense of Place | The cultural built heritage makes the town special and provides a sense of home to its inhabitants. |
| 2. Society | The cultural built heritage of the town is a source of pride for the inhabitants, who together promote its preservation and use. |
| 3. Urban fabric, Structure, and Spaces | The historic centre plays a crucial role in the daily life of the inhabitants, hosting the most important administrative, social, cultural, religious, and commercial functions, as well as offering attractive housing, work, and public spaces. |
| 4. Services and Facilities | The cultural built heritage is widely used by all age groups for leisure, cultural, and educational activities; these are often related to local traditions and festivities. |
| 5. Economy | The cultural built heritage of the town is relevant to the local economy by offering job opportunities in the construction, tourism, and event sectors. It helps attract business and investment. |

Source: Our own elaboration.

On the basis of the research carried out, it has been possible to achieve the main objective, which was to develop the authors' proposal of 20 indicators that, in the authors' opinion, comprehensively cover the problem of the impact of cultural heritage on the quality of life in cities, with particular emphasis on small and medium-sized towns. In the proposed set, four indicators have been assigned to each of the five dimensions in which CBH can affect the quality of life.

## 4. Results

### 4.1. Inadequacy of Existing Indicators—Results from Literture Review

The first research objective was to find an answer to the question of whether CBH is taken into account in studies of indicators of quality of life and, if so, what indicators are used to determine these relationships. For this purpose, scientific and strategic documents relating mainly, but not exclusively, to the area of interest to us (Poland and Germany) were analysed. Of this large pool, 22 documents were selected that offered a clearly indicator-based approach to QoL research (see Table A1 in Appendix A).

In these studies, we identified 412 QoL indicators. These indicators measured QoL both objectively and subjectively. Thirty-seven were found to be directly or indirectly related to cultural heritage, with a particular emphasis on CBH (Table 2). This represents 9.0% of all the indicators proposed in the 22 reviewed studies. Only eight indicators were directly related to cultural heritage, which represents 2.0% of the entire pool of 412 indicators (and 21.6% of the 37 selected indicators), and the remaining 29 (7.0% of the 412 indicators or 78.4% of the 37 indicators) we found to be only indirectly related. This literature review analysis confirms our preliminary thesis that cultural heritage per se, and CBH in particular, have been marginalised in QoL studies.

**Table 2.** The indicators directly and indirectly related to cultural heritage in 22 analysed studies.

| No. | Dimensions [1] | Name of Indicator |
|---|---|---|
| 1. | 1 | ● Satisfaction with the town (S4, S8, S10, S12, S17) [2] |
| 2. | 1 | ● Social order: the feeling of being at home—sense of being lost (S22) |
| 3. | 1 | ● Personal connection to the town (S5) |
| 4. | 1, 2 | ● Local bonds to the place (relatives, friends, house, landscape) (S10) |
| 5. | 1, 2 | ● Reputation of the neighbourhood (S10) |
| 6. | 1, 3 | ● **Image of the town/skyline (recognisability and symbolic value) (S6, S14)** [3] |
| 7. | 1, 2, 3 | ● Satisfaction with the nearby environment (S10) |
| 8. | 1, 2, 3 | ● Satisfaction with the neighbourhood (S10) |
| 9. | 2 | ● Number of local NGOs (S1, S7) |
| 10. | 2 | ● Close/attractive/collaborative town for citizens (S6, S13) |
| 11. | 2 | ● Forms of public participation (known, desired, used) (S8) |
| 12. | 2, 3, 4, 5 | ● Perception of the town (S8) |
| 13. | 3 | ● Mixture of commerce, recreational offerings and residential function (S11) |
| 14. | 3 | ● **Architectural quality and degree of maintenance of public areas in public housing (S14)** |
| 15. | 3 | ● Quality and level of maintenance of public-housing buildings (S14) |
| 16. | 3 | ● Architectural quality (street furniture, art installations) of open spaces (S14) |
| 17. | 3 | ● **Adaptation of the type and architectural character of the building to the local climate (S14)** |
| 18. | 3 | ● **Overall impact of colour and building harmony (materials, paving, openings, proportions) (S14)** |
| 19. | 3 | ● Urban and architectural form and arrangement: compact—dispersed; uniform—chaotic; existence of street furniture—lack of street furniture (S22) |
| 20. | 3 | ● Functional order: good social infrastructure—lack of social infrastructure (S22) |
| 21. | 3 | ● Aesthetic order: beautiful—unattractive (S22) |
| 22. | 3 | ● Functional order: saturation with social infrastructure—lack of saturation with social infrastructure (S22) |
| 23. | 3 | ● Satisfaction with the offer of goods in neighbourhood (S8) |
| 24. | 3 | ● Distance to workplace in km (S10) |
| 25. | 3 | ● Duration of commute to work (S10) |
| 26. | 3 | ● Facilities within walking distance (S11) |
| 27. | 3, 5 | ● Presence of shops (S4, S6, S12) |
| 28. | 3, 5 | ● Local offers of food/clothing/household goods/other (city centre, district, other city, online) (S8, S11) |
| 29. | 3, 5 | ● Density of bars and restaurants (S8, S11) |
| 30. | 4 | ● **Satisfaction with cultural organisations (S4, S6, S12)** |
| 31. | 4 | ● **Number of museums (S7)** |
| 32. | 4 | ● **Music events (S8)** |
| 33. | 4 | ● Markets, city fairs, and nightlife (S8) |
| 34. | 4 | ● Local recreation possibilities (S11) |
| 35. | 4 | ● **Density and relevance of the museum heritage (museums, archaeological sites, monuments) (S14)** |
| 36. | 5 | ● Role of tourism in the city (S6) |
| 37. | 5 | ● Influence of tourism on quality of life (profit for retailers, support of regional culture, more services offered) (S6) |

[1] The descriptions of dimensions 1–5 can be found in Table 1. [2] The numbers in brackets refer to the analysed studies listed in Table 2. [3] The indicators in bold type are directly related to cultural heritage. Source: Our own elaboration.

Furthermore, it was observed that even when CBH is included in the assessment of the quality of life, it is usually limited to the assessment of its spatial-environmental aspects. At this point, an important research question was raised: What are the other aspects of QoL that can be supported by the CBH component? An analysis of a wide range of QoL indicators allowed for the designation of five spheres in which QoL is usually assessed: urban identity/sense of place; society; urban fabric, structure, and spaces; services and facilities; and economy. This division has been discussed and approved by the selected experts from the fields of QoL research and statistics, SMT research, and urban renewal and conservation.

Having decided on the divisions, we confronted the five spheres with 37 previously extracted indicators related to CBH in order discern the sphere of the scientific literature in which their potential is usually perceived. Cross-analysis confirmed our observation: most of the selected indicators (21) relate to the third dimension of QoL, namely urban fabric, structure, and space. The other four dimensions were represented by a significantly smaller number of indicators (ranging from six to eight). At the same time, we confirmed

that cultural heritage is marginalised in research on the quality of life, and this particularly applies to CBH.

### 4.2. Cultural Built Heritage Matters—Results from the Focus Group Discussions

The assumption that CBH influences all five identified dimensions of QoL was the basis for the research in the subsequent stages. The existence of multifaceted interdependencies between CBH and the QoL in SMTs was confirmed by the focus group discussions in the 10 pilot towns. The results are classified along five dimensions, as presented below. The findings are supported by references to relevant literature.

#### 4.2.1. Urban Identity/Sense of Place

The assumption behind the first dimension emphasizes the potential of CBH to make a town special and to help "define the character of the place" ([25], p. 1). Heritage not only contributes to the shaping of everyday life in SMT, but it is also an important factor in improving the attractiveness of these settlements [25,79]. At the same time, the individuality, spatial/visual quality, and coherence of the historic fabric, along with its human scale and originality, foster emotional ties between the local population and their urban environment [25]. Memories and meanings are created through associations and interactions between individuals and the built fabric [25,80,81]. The CBH not only helps to meet the aesthetic needs of a society [82] but also has an immense impact on individual and collective identities [25] and civic pride in a town.

The degree of local identification with the CBH and the level of individual attachment varies from town to town, depending on local factors. Participants in the focus group meetings stressed the existing attachment to a place and the identification with it, especially where the unique values of tangible and intangible heritage were high. The sense of belonging to the town was also strengthened by the attractiveness of the surrounding countryside. A positive external perception of a town, i.e., through the opinions of visitors, tourists, and businesses, may also affect the sense of local identity and attachment to the place.

There was also general agreement among the focus groups that cultural heritage, whether tangible or intangible, makes the town unique. Some participants stated that the sense of a town's uniqueness is influenced by a clear urban structure, the small scale, and a good sense of orientation. The beauty of the town itself was also mentioned as a factor that improved its uniqueness. In four out of ten towns, the focus group participants complained that there were too few high-quality meeting places in the old town space. The availability of such meeting places not only helps enhance social integration, but also facilitates the creation of emotional bonds and attachment to the CBH [83,84].

History may be an important factor influencing the individual perception of local CBH, particularly in Lower Silesia, where, after World War II, large-scale migration drained towns of their native populations, thereby disrupting the traditional social identity. In such cases, any acceptance of the CBH, which is largely the heritage of "others", may take longer to achieve.

#### 4.2.2. Society

The assumption made for the second dimension states that the CBH of SMTs boosts local pride while supporting social engagement and cohesion [25,79,85]. The literature review confirmed this assumption by underlining the potential of CBH to create social capital [79,86] and social cohesion [25,79,85]. Based on research in the United States, Oliver [87] found that social capital as measured by community involvement tends to be higher in smaller cities.

Compact spatial structures, which often occur in SMTs with a distinctive and well-preserved historic centre, favour more direct communication between local authorities and socially engaged citizens than is the case in more loosely structured settlements [88,89]. For this reason, SMTs often enjoy a good level of social participation, active citizenship,

strong formal and informal social networks, and intense cooperation between stakeholders involved in various urban policy issues [90,91].

According to the literature, the CBH is (at least in part) a source of pride for the citizens, which has been confirmed by our focus group meetings. The discussions also made it clear that this relationship depends not only on local circumstances but also on several external factors. In general, we can say that the overall economic and social situation of the town directly affects the condition of the CBH, but also indirectly influences the level of local pride.

In eight of the ten towns analysed, the focus groups discussions revealed that civic engagement in preservation of built urban heritage is observed at various levels, thus confirming the thesis that "urban conservation offers manifold opportunities for civic engagement" ([92], p. 19). However, it should be noted that the involvement of residents is often heterogeneous and fragmented. In addition, only a small number of people regularly participate in initiatives to preserve historic buildings. Therefore, the impact of cultural heritage on local participation and social cohesion is rather limited.

### 4.2.3. Urban Fabric, Structure, and Space

The assumption assigned to the third dimension relates to the positive impact that the old-town area can have on the lives of local residents in view of its multifunctionality, visual attractiveness, and the number of public spaces. In many SMTs, the historic centre makes up a larger proportion of the total settlement area than in larger cities. Clearly, this historic core is often of key importance in daily life, especially in smaller towns [88]. Frequently, it is the hub of basic administrative, social, cultural, religious, and commercial functions [59]. In addition, the compact size of SMTs facilitates their management and the daily functioning of the inhabitants [93].

The activation and appropriate development of old-town areas may help reduce suburbanisation, which can lead to a range of spatial, social, and economic problems. Compact, densely populated areas are easier to provide with public transportation. The cost of building essential urban infrastructure is also lower. In general, the compact urban structure of SMTs is considered a vital contributing factor to QoL by promoting interaction and social cohesion, facilitating orientation and navigation in the town, and ensuring easy access to natural areas [93–95].

While the historic centres of all analysed towns were found to host important administrative, social, cultural, religious, and commercial functions, these differed in the degree and form of their developmental dynamics. In particular, the size of the town and its economic health appeared to affect the level and range of activities and functions located in the historic centres. The functions of the centres naturally tend to be more diverse in bigger and economically stronger towns than in smaller ones. Furthermore, the urban structure and historic landmarks were confirmed by the focus groups to ease navigation and orientation in all of the Polish towns.

To varying degrees, all town centres are struggling with a loss of functions, partly due to suburbanisation processes. In most cases, commercial amenities for daily life, as well as religious and administrative facilities, are located in the historic centres. Participants in the focus group discussions repeatedly mentioned that other important public, social, cultural, and commercial functions—such as courts, schools, kindergartens, health services, sports facilities, theatres, and supermarkets—are often located outside the town centres or are currently relocating elsewhere. A problem often pointed out during the meetings was the lack or low quality of existing public spaces in the old-town areas.

All town centres are used for residential purposes, including historic buildings. In four of the SMTs examined, living conditions in the historic centre were explicitly described as positive, despite certain shortcomings related to traditional urban structure/fabric, such as a lack of greenery and daylight in the houses, outdated floor plans, comparatively high rents, or noise during weekends or public events.

### 4.2.4. Services and Facilities

Closely linked to the assumption made in the third dimension, the assumption made in the fourth dimension focuses on whether the CBH in the pilot towns adds to the quality of urban life by providing leisure, cultural, and educational services or facilities used across various age groups. Our literature review suggested that CBH generally has a high potential in this regard [79]. In particular, this assumption is related to the idea that the cultural life of SMTs tends to be significantly shaped by civic associations and typically includes traditional festivities [96].

There is a considerable disparity in the extent to which the individual historic centres and buildings are used for leisure, cultural, and educational activities. However, participants in almost all focus groups mentioned a general lack of liveliness and a desire for more activities in the urban centres. This suggests that the potential of the old towns as a venue for leisure, cultural, and educational activities related to the local and regional heritage is not fully exploited in any of the towns.

All historic centres host permanent or regular cultural facilities and services. Festivities and events such as the celebration of holidays, town galas, and thematic fairs seem to be organised in the historic centre of all investigated towns, and the old buildings serve as a scenic backdrop. On the other hand, the potential of the historic city centre as a place for spontaneous meetings and free-time activities can be described as lost or at least not fully exploited. In most of the studied cases, the centres only come to life during larger, organised events such as town galas. There is some disparity in the degree to which the local cultural heritage is used for educational purposes. While most towns host facilities such as museums in the old centre, other educational facilities were reported to be located (mainly or partly) outside the old town.

### 4.2.5. Economy

The assumption made in the fifth dimension addresses the potential of cultural heritage as an economic factor in SMTs. Based on the literature review and opinions expressed at the focus group discussion, we found that cultural heritage can certainly play a significant role in economic sectors such as tourism, construction, local crafts, and culture. In many cases, it is fundamental in attracting tourists. Additionally, highlighting local traditions and craftsmanship may be an important element in the marketing strategies of various businesses that continue to produce goods related to the local cultural heritage. Conservation and protection of the architectural heritage can also create employment opportunities in the construction and crafts sector.

Despite all the potential economic benefits of cultural heritage in SMTs, historic urban fabric can also be a burden with respect to funding needed for its preservation. The modernisation and adaptation processes relating to historic buildings are often limited by the requirements of heritage protection policies and regulations.

In recent years, global trends have helped to increase the attractiveness of SMTs as a place of residence. For example, due to the greater opportunities for remote work, many people are able to move to smaller towns without losing their well-paying jobs in larger urban centres [93,97]. This growing appeal of SMTs may also be influenced by lifestyle trends, housing preferences, and the lifestyle trends of younger generations (millennials). For example, various media reports have documented this development in the United States. More and more young people are perceiving the benefits of living in smaller centres, which include lower living costs and a greater sense of community [98–100].

The assumption drawn for the economic potential of the CBH applies to the towns analysed in varying degrees. The potential seems to be realised with more success in larger settlements. Based on meetings with local groups, it can be concluded that not only architectural objects, but also elements of intangible heritage, such as local crafts, can contribute to the economy. For most of the towns involved in the project, tangible or intangible cultural heritage is the basis for tourism.

In six out of ten towns analysed, participants at the meetings mentioned negative economic phenomena related to the disappearance of commercial activities in historic centres. This is due, among other factors, to the construction of large retail parks on the outskirts of the settlements. Clearly, this reduces the importance of the old-town centre for everyday activities and may drive up vacancies in the building stock and even cause a significant economic marginalisation of the historic core with its traditional market square. Furthermore, different legal, financial, or structural barriers can have a negative impact on the exploitation of the economic potential of cultural heritage.

### 4.3. Proposed Indicator-Based Approach

Due to the difficulty in obtaining a comprehensive and structured set of indicators measuring the links between CBH and the quality of life in towns (with a particular emphasis on SMTs), the authors determined to prepare a new set of indicators to reflect the multifaceted influence of CBH on quality of life in a more balanced way (Table 3). The selected indicators from the first set provided a starting point, but since most only indirectly referred to quality of life in the context of CBH, they required some transformation and supplementation. In the process of developing the indicators, the results of the focus group meetings played a crucial role.

**Table 3.** The proposed set of quality of life indicators in the context of cultural heritage.

| Dimension | Name of Indicator | Type of Indicator | Data Sources |
|---|---|---|---|
| 1 | The level of identification with the town | subjective | survey research |
| 1 | Sense of pride in the town | subjective | survey research |
| 1 | Visual attractiveness of the urban space | subjective | survey research |
| 1 | The level of uniqueness of the architectural cultural heritage of the town | subjective | survey research |
| 2 | Sense of local community | subjective | survey research |
| 2 | Opportunity to get involved in the local community life | subjective | survey research |
| 2 | Sense of having influence on the situation in the town | subjective | survey research |
| 2 | Number of institutions, organisations and associations related to the cultural heritage per 10,000 inhabitants | objective | municipal data, public statistics, field research, register of entrepreneurs |
| 3 | Land use mix of the town centre | objective | field research, spatial databases |
| 3 | Percentage of residential buildings in the town centre with retail and service facilities on the ground floor | objective | field research |
| 3 | Bicycle and pedestrian facilities in the town centre | objective | field research, spatial databases |
| 3 | Satisfaction with public spaces in the town centre | subjective | survey research |
| 4 | Satisfaction with the cultural offerings in the town | subjective | survey research |
| 4 | Number of cultural events that take place in the town centre during the year per 10,000 inhabitants | objective | municipal data, interviews with local stakeholders, public statistics |
| 4 | Participation of residents in cultural events taking place in the town centre | objective | survey research |
| 4 | Number of heritage/listed buildings used for cultural, social, or educational amenities | objective | municipal data, field research, spatial databases |
| 5 | Impact of cultural heritage on the economic development of the town | subjective | survey research |
| 5 | Number of enterprises based on local cultural heritage (tangible and intangible) per 10,000 inhabitants | objective | municipal data, field research, public statistics, register of entrepreneurs |
| 5 | Vacancy rate in the town centre | objective | field research, spatial databases |
| 5 | Number of tourists visiting the town per 10,000 inhabitants (annually) | objective | municipal data, public statistic |

Source: Our own elaboration.

The aim was to create a fairly compact set of indicators to comprehensively evaluate the five dimensions mentioned above. Four indicators were assigned to each of the five proposed dimensions in which quality of life and cultural heritage are interrelated. This created a complete set of 20 indicators. As the objective was to capture the analysed phenomena on a multifaceted basis, the set includes 10 objective and 10 subjective indicators.

However, it should be noted that the individual dimensions are captured disparately by subjective and objective indicators. Due to their specificity, half of the indicators can only be assessed by means of a survey. We proposed adopting a uniform 5-point response scale for all questions (i.e., 1—lowest rating; 5—highest rating). Other data sources for some indicators could be obtained from municipal or town offices, public statistics, field studies, registers of entrepreneurs, and spatial databases. Although here the authors applied the described set of indicators to SMTs, it can also be usefully employed to analyse larger urban centres. The proposed set of indicators is suitable for conducting comparative research in two different spatial approaches:

— between cities of the same size, with and without a well-preserved CBH;
— between cities of different sizes, both with a well-preserved CBH.

The first dimension concerning urban identity/sense of place deals with the personal relationship between residents and their town and the individual perception of its value. All four indicators assigned to this dimension are subjective. The first two focus on the bond that can be formed between citizens and their town, resulting in a strong identification with it and a sense of individual pride in being a resident. The next two indicators address the question of whether tangible cultural heritage can satisfy the aesthetic needs of the inhabitants and the feeling of uniqueness of the urban scenery created by, among other things, the architectural heritage of the town.

The indicators assigned to the second dimension address the impact of cultural heritage on the formation of social bonds and social capital. Three subjective indicators and one objective indicator are used to describe this dimension. The first indicator addresses the sense of connection of a resident with other locals and thus the creation of a town community. These bonds can be strengthened by shared goals, values, and activities related to cultural heritage. The next two indicators are related to opportunities to get involved in the local community and the resulting influence on the level of local activities. The final indicator attributed to the second dimension refers to the number of organisations, institutions, and associations related to the local cultural heritage that operate in the town. Through their work, they can contribute to the growth of local social capital and thus constitute an important link in the protection, management, and dissemination of knowledge about the cultural heritage of the town.

Three objective indicators and one subjective indicator were assigned to the third dimension, relating to urban fabric, structure, and space. The first two of these indicators address the issue of the multifunctionality of old-town areas, including the availability of retail and service facilities. A multifunctional spatial structure with numerous retail and service facilities allows residents of old-town areas to meet many of their daily needs in their local neighbourhood. The third indicator concerns the presence of pedestrian and bicycle infrastructure (in the form of pavements and bicycle paths) in the historic centre. This indicator can be expressed as the density of a particular type of infrastructure, or the ratio of the length of pavements, bicycle paths, and shared spaces to public roads. At a time of global climate change and the pressing challenges this brings to society, sustainable urban mobility is clearly an important feature of the city of the future. In this regard, compact SMTs bring benefits by allowing residents to cover comparatively short distances on foot or by bicycle. The final subjective indicator determines the level of the citizens' satisfaction with public spaces located in the old-town area. These spaces play an important role in human interactions, especially today, when many spaces have been privatised and commercialised.

Three objective and one subjective indicators were also assigned to capture the fourth dimension in terms of services and facilities. The first one concerns the satisfaction of residents with the local cultural offerings. In this case, cultural services are considered as a whole and not just as those focusing on cultural heritage. The next two indicators are related to the number of cultural events held in the city centre per 10,000 inhabitants, and the level of participation of the residents in these events. These three indicators assess the cultural offerings in the town both quantitatively and qualitatively. Furthermore, they refer to the entire set of cultural offerings, as the local cultural heritage can also contribute

to the organisation of events not directly related to it. For example, visually attractive old-town space can be an important element in attracting outdoor cultural events that are not necessarily related to local cultural heritage. The aim of the last indicator in this group is to determine the extent to which local historic buildings and structures are actually used for various purposes. This better reflects their potential than would a total number of registered monuments, regardless of the role they play in urban life.

Regarding the fifth dimension of the relationship between cultural heritage and the economy, the authors also proposed one subjective and three objective indicators. The subjective indicator is designed to assess the impact of cultural heritage on the economic health of the city, as viewed by its inhabitants. The second, concerning the strength of the impact of local heritage on entrepreneurship, is simply the number of enterprises based on local cultural heritage (tangible and intangible) per 10,000 inhabitants. The next indicator shows the degree to which the historic centre currently retains its role as a commercial and service centre for the town as a whole (vacancy rate). The last indicator is the number of tourists per 10,000 inhabitants, which measures the tourist potential of the town.

## 5. Conclusions and Prospects

The review of the existing literature related to quality of life reveals the need to better recognise and instrumentalise the relationship between quality of life and cultural built heritage. The underrepresentation of built cultural heritage in quality-of-life research may be caused by the poor availability of adequate data, but also by poor recognition of the mutual relationships between these spheres.

Departing from two assumptions: first, that cultural heritage may be an important factor influencing the quality of life, and second, that this influence is particularly noticeable in small and medium-sized towns, this article makes an attempt to provide a universal set of indicators for measuring the level of mutual interrelation between QoL and CBH.

The paper demonstrates that the impact of built cultural heritage on the quality of life in small and medium-sized towns can be assessed on many levels. Our study proposes five dimensions through which the interrelationship between QoL and CBH can be observed. The strength of these links varies from place to place and depends on various contextual factors, such as the condition of the heritage in a town, the institutional framework, and the appropriation of the heritage by the local population [101]. The aim of this study was to underpin the five dimensions with a balanced set of indicators evaluating the quality of life in historic SMTs. The multidimensional nature of the relationships between QoL and CBH requires both subjective and objective indicators to adequately capture heritage-related QoL.

The proposed set of indicators has some limitations. Firstly, the use of two types of indicators allows for a more comprehensive presentation of the analysed issues, but it also may serve to complicate the interpretation of obtained results. Moreover, subjective indicators may be difficult to use in practice because general awareness of the CBH is rather low and potential respondents may find some questions ambiguous. Secondly, the compactness of the set (20 indicators) increases its usefulness but may present an incomplete picture, since some aspects may not have been taken into account. Thirdly, the offered set of indicators has been tailored specifically for small and medium-sized towns. It may seem hardly applicable for larger cities.

It must be mentioned as well that although we aspire to offer a universal tool for the measurement of QoL and CBH relations, we are fully conscious that our study is strongly grounded in the very specific socio-cultural context of Central Europe, and the proposed set of indicators may not answer the required specificity of different cultures and locations. An important future line of research would be to conduct a pilot study to test the proposed set of indicators. It is vital that the relationship between cultural heritage and quality of life is more widely recognized and mainstreamed in research, both by improving and developing existing indicators used to assess quality of life and well-being, but also by developing new ones.

**Author Contributions:** Conceptualization, S.K., M.B., W.M., K.G., M.K. and R.K.; methodology, S.K., W.M., K.G., M.K., E.B.-S. and S.A.-A.; software, S.K.; validation, S.K., K.G., M.K., E.B.-S. and S.A.-A.;

formal analysis, S.K., M.B., K.G., M.K., E.B.-S., B.K. and S.A.-A.; investigation, S.K., M.B., K.G., M.K., E.B.-S., B.K. and S.A.-A.; resources, S.K., K.G., M.K., E.B.-S., B.K. and S.A.-A.; data curation, S.K., K.G., M.K., E.B.-S., B.K. and S.A.-A.; writing—original draft preparation, S.K., M.B. and W.M.; writing— review and editing, S.K., M.B., W.M., K.G., M.K., E.B.-S. and B.K..; visualization, S.K. and E.B.-S.; supervision, M.B. and R.K.; project administration, R.K., E.B.-S., B.K. and S.A.-A.; funding acquisition, R.K. and B.K. All authors have read and agreed to the published version of the manuscript.

**Funding:** The study presented in this paper was carried out within the project "REVIVAL!-Revitalisation of the historic towns in Lower Silesia and Saxony", which was funded within the programme IN-TERREG Polska-Sachsen 2014–2020 by the European Regional Development Fund (ERDF) of the European Union.

**Acknowledgments:** We would like to thank Marzenna Halicka-Borucka, Magdalena Pietrukiewicz, Przemysław Malczewski, Steve Naumann, and the research assistants Marek W. Jaskólski, Jule Morgenroth, Claudia Romelli, Domenik Vogt, Iryna Yaniv, and Johannes Hübner, who were involved in the project.

**Conflicts of Interest:** The authors declare no conflict of interest.

## Appendix A

**Table A1.** The list of studies covered by the analysis.

| Code | Study |
|---|---|
| S1 | ZDF Deutschland-Studie 2018 [ZDF Study on Germany 2018]. Available online: https://www.prognos.com/de/projekt/zdf-deutschland-studie-2018 (accessed on 24 November 2021) [61] |
| S2 | Bundeskanzleramt, Stab Politische Planung, Grundsatzfragen und Sonderaufgaben und Bundesministerium für Wirtschaft und Energie, Leitungs- und Planungsabteilung. *Bericht der Bundesregierung zur Lebensqualität in Deutschland [Report of the German Government on the Quality of Life in Germany]*; Presse- und Informationsamt der Bundesregierung: Berlin, Germany, 2016. [62] |
| S3 | Büttner, T.; Ebertz, A. Lebensqualität in den Regionen: Erste Ergebnisse für Deutschland [Quality of Life at the Regional Level: First Results for Germany]. *ifo Schnelldienst* **2007**, *60*, 13–19. [63] |
| S4 | Lemcke, J.; Zakrewski, T. *Dresden in Zahlen [Dresden by Numbers]*; Landeshauptstadt Dresden: Dresden, Germany, 2010. [64] |
| S5 | Hietzgern, K. *Partizipation, Identifikation und Lebensqualität im städtischen Raum. Eine empirische Studie in der niederösterreichischen Stadt Krems [Participation, Identification, and Quality of Life in Urban Areas. Empirical Study in the Town of Krems in Lower-Austria]* (MSc Thesis); Universität Wien: Vienna, Austria, 2009. [16] |
| S6 | Fachhochschule Westküste. Institut für Management und Tourismus (IMT). *Tourismus und Lebensqualität in Cittaslow-Städten. Studienergebnisse in den Cittaslow-Städten Bad Essen, Deidesheim und Meldorf [Tourism and Quality of Life in Cittaslow-Cities. Study Results in the Cittaslow-Cities Bad Essen, Deidesheim and Meldorf]*; Fachhochschule Westküste Institut für Management und Tourismus: Heide, Germany, 2018. [17] |
| S7 | Schmitz-Veltin, A.; West, C. *Leben und Arbeiten in Penzberg: Studie zur Lebensqualität [Living and Working in Penzberg: Study on the Quality of Life]*; Universität Mannheim: Mannheim, Germany, 2006 [18] |
| S8 | Heiden, A. *Lebensqualität in Siegen. Eine Studie zum Stadterleben der Siegner Bürgerinnen und Bürger [Quality of Life in Siegen. A Study on Perceptions of the City Among the Citizens of Siegen]*; Stadt Siegen: Siegen, Germany, 2009. [19] |
| S9 | iW Consult. *Wirtschaftswoche Städteranking 2018 [Wirtschaftswoche's Ranking of Cities 2018]*. Available online: https://www.wiwo.de/politik/deutschland/staedteranking/definitionen/ (accessed on 24 November 2021). [65] |
| S10 | Sturm, G.; Walther, A. *Landleben-Landlust? Wie Menschen in Kleinstädten und Landgemeinden über ihr Lebensumfeld urteilen [Country Life—Desire for the country? How People in Small Towns and Rural Communities Evaluate Their Living Environment]*, BBSR-Berichte KOMPAKT, 10/2010; Bundesamt für Bauwesen und Raumordnung: Bonn, Germany 2010. [20] |
| S11 | Sturm, G., Walther, A. *Lebensqualität in kleinen Städten und Landgemeinden: Aktuelle Befunde der BBSR-Umfrage [Quality of Life in Small Towns and Rural Communities: Current findings of the BBSR-Survey]*, BBSR-Berichte KOMPAKT, 5/2011; Bundesamt für Bauwesen und Raumordnung: Bonn, Germany, 2011. [21] |
| S12 | European Commission. *Quality of life in cities. Perception survey in 79 European cities*; Publications Office of the European Union: Luxembourg, Luxembourg, 2013. [66] |
| S13 | Organisation for Economic Co-operation and Development (OECD). *OECD Better Life Initiative Country Notes Germany*. Available online https://www.oecd.org/statistics/Better-Life-Initiative-country-note-Germany.pdf (accessed on 24 November 2021). [67] |
| S14 | Istituto nazionale di statistica (Istat). *Il benessere equo e sostenibile in Italia [Fair and Sustainable Wellbeing in Italy]*; Istat: Roma, Italy, 2017. [68] |
| S15 | Grifoni, R.C.; D'Onofrio, R.; Sargolini, M. Quality of Life in Urban Landscapes: In Search of a Decision Support System. Springer: Berlin/Heidelberg, Germany, 2018. [69] |

**Table A1.** *Cont.*

| Code | Study |
|------|-------|
| S16 | Główny Urząd Statystyczny (GUS). *Jakość życia w Polsce [Quality of Life in Poland]*; Główny Urząd Statystyczny: Warszawa, Poland, 2013. [70] |
| S17 | Polityka. *Ranking jakości życia [Quality of Life Ranking]*. Available online: https://www.polityka.pl/tygodnikpolityka/mojemiasto/1762774,1,ranking-jakosci-zycia.read (accessed on 24 November 2021) [71] |
| S18 | Bański, J.; Pantyley, V. Warunki życia we wschodniej Polsce według regionów i kategorii jednostek osadniczych [Living Conditions in Eastern Poland by Region and Settlement Units]. *Nierówności społeczne a wzrost gospodarczy* **2013**, *34*, 107–123. [72] |
| S19 | Jeran, A. Jakość i warunki życia—perspektywa mieszkańców i statystyk opisujących miasta na przykładzie Bydgoszczy, Torunia i Włocławka [Quality And Conditions of Life—Perspective Residents and Statistics Describing the Town on the Example Bydgoszcz, Torun and Wloclawek]. *Roczniki Ekonomiczne Kujawsko-Pomorskiej Szkoły Wyższej w Bydgoszczy* **2015**, *8*, 222–235. [73] |
| S20 | PricewaterhouseCoopers (PwC). *Raport na temat wielkich miast Polski [Report on the Great Cities of Poland]*. Available online: https://www.pwc.pl/pl/sektor-publiczny/raporty_warszawa-pol.pdf (accessed on 24 November 2021). [74] |
| S21 | Binda, A.; Łobodzińska, A.; Motak, E.; Nowak-Olejnik, A.; Jarząbek, B.; Poniewierska, A. *Miasta województwa małopolskiego: zmiany, wyzwania i perspektywy rozwoju [Cities of the Malopolska Region: Changes, Challenges and Development Perspectives]*; Małopolskie Obserwatorium Rozwoju Regionalnego, Departament Polityki Regionalnej, Urząd Marszałkowski Województwa Małopolskiego: Kraków, Poland, 2018. [75] |
| S22 | Konecka-Szydłowska, B. Ocena przestrzeni publicznej małych miast aglomeracji poznańskiej [Assessment of Public Spaces of Small Towns in the Poznań Agglomeration]. *Problemy Rozwoju Miast* **2016**, *3*, 5–12. [76] |

Source: Our own elaboration.

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
