# Peer review of "Using Indicators to Evaluate Cultural Heritage and the Quality of Life in Small and Medium-Sized Towns: The Study of 10 Towns from the Polish-German Borderland"

_sustainability, doi:10.3390/su14031322_

Round 1

Reviewer 1 Report

The paper ”Using indicators to evaluate cultural heritage and the quality of 
life in small and medium-sized towns. The study on 10 towns from Polish-German borderland.” brings to our attention the results of research focused on the interdependencies between built cultural heritage and the quality of life in small and medium-sized towns.

Although part of the results of the paper is based on the literature review, the authors complete the research with with empirical research, conducted in ten SMTs in the border area between Poland (Lower Silesian and Lubusz Voivodeships) and Germany (Saxony). The bibliographic references are numerous and conclusive for the approached subject. A concise and well-organized work.

Please insert better resolution versions for the figures 1 and 2. These images are blurred.

Author Response

Thank you very much for your positive evaluation of our article. We hope that the new version, revised according to the comments of the other Reviewers, will also be satisfactory to you.

Reviewer 2 Report

First of all I would like to thank contributors for choosing such an important issue. However, from the academic point of view I have mentioned my observations on the paper in the following sections.

1. Should revise and give emphasis on nature of research, analytical tools, techniques with results and and implication.Sentence in the abstract should be started in line of title. There is need to have both theoretical and and practical application in the abstract which is bit absent. There is no meaningful objective which is supported by evidence of theoretical ground. Abstract must contain: (a) Originality of the research; (b) Research objective; (c) Method; (d) Empirical result; (e) Practical implications. The abstract must be structured.

2. Introduction section lacks clarity and main motivation of the study is missing. Need to provide background information and set the context about your study in the context of Polish-German, Introduce the specific topic of your research and explain why it is important, Mention past attempts to solve the research problem or to answer the research question and conclude the Introduction by mentioning the specific objectives of your research. The reviewer bit afraid that the author does not use the appropriate literature support and even has not mentioned the main outcome of the study.

3. As a part of this section you can be included more relevant, recent and updated theoretical foundation of this study, lastly include a research gap of your study. The author should critically review the literature by highlighting what is missing in the literature and the weaknesses that need to be addressed (the research gap) rather than the descriptive manner/ just reporting the general concept of the topic.

4. Specify methods and procedures of your study very clearly, as part of this section you can include research design, variables covered, data analysis method etc. The article does not explain the models used. Theoretical framework is not covered. The diagram is unclear and not description is missed.  What is the reason/background to choose these 5 hypotheses? 5 different hypotheses were given methods section which doesn't provide enough justification for studying this phenomenon. And the hypothesis made by the author could not be tested properly. Thus the authors should mention exact methodology that has been followed with support from the literature.

5. No detailed discussion in the comparison among reviewed studies and showing the differences and incremental academic contributions between current study and them. This section lack discussion in the connections between existing studies and the current one. Therefore, it is not clear the academic contributions of the current study to the literature. Prove the results and find a linkage among results, research questions and hypothesis by using previous studies/literature review, whether your hypothesis accepted or rejected, matches with similar results and dissimilar. If dissimilar, mention what cause.

6. The conclusion must shows the answer from the research objective and the policy implications and as a part of this section you can be concluded theoretical implication, future implication, and recommendation for future practice briefly. Need to include limitations of the study in this section.  

BEST OF LUCK!

Author Response

Thank you very much for your in-depth and inspiring comments. The manuscript has been thoroughly revised and we hope it will now better reflect and explain the idea of the research. Below we provide detailed responses to the comments.  

Point 1. Should revise and give emphasis on nature of research, analytical tools, techniques with results and and implication. Sentence in the abstract should be started in line of title. There is need to have both theoretical and and practical application in the abstract which is bit absent. There is no meaningful objective which is supported by evidence of theoretical ground. Abstract must contain: (a) Originality of the research; (b) Research objective; (c) Method; (d) Empirical result; (e) Practical implications. The abstract must be structured.

Response 1: The abstract has been entirely rewritten and restructured according to the suggestion. We hope that it now better reflects the theoretical and practical aspect of the study undertaken and also highlights the objective of the study. The abstract includes information about the research methods used, which is further developed in Chapter 3: Research Methods. It has been entirely re-edited and given a clearer structure.     

Point 2. Introduction section lacks clarity and main motivation of the study is missing. Need to provide background information and set the context about your study in the context of Polish-German, Introduce the specific topic of your research and explain why it is important, Mention past attempts to solve the research problem or to answer the research question and conclude the Introduction by mentioning the specific objectives of your research. The reviewer bit afraid that the author does not use the appropriate literature support and even has not mentioned the main outcome of the study.

Response 2: The Introduction part was thoroughly rewritten. An attempt was made to better present both the motivation for taking up the topic of the study and the context of the German-Polish border region. As suggested, the specific objectives of the research were also emphasized the objectives of research and the research gap we attempt to fill by our study

Point 3. As a part of this section you can be included more relevant, recent and updated theoretical foundation of this study, lastly include a research gap of your study. The author should critically review the literature by highlighting what is missing in the literature and the weaknesses that need to be addressed (the research gap) rather than the descriptive manner/ just reporting the general concept of the topic.

Response 3: We have significantly modified the Literature review section and changed its title to: “CBH as a parameter of quality of life in small and medium sized towns”. We have tried to improve and enrich the discussion basing on the current literature and indicate the research gap. We have also decided to better focus the track of discussion, by reducing the consideration of the problem of small and medium-sized cities. We would like to point out that the literature review in this section is not the only one - in the "results" section we refer in more detail to the 22 thematic studies that were considered most important for the topic of our research.

Point 4. Specify methods and procedures of your study very clearly, as part of this section you can include research design, variables covered, data analysis method etc. The article does not explain the models used. Theoretical framework is not covered. The diagram is unclear and not description is missed.  What is the reason/background to choose these 5 hypotheses? 5 different hypotheses were given methods section which doesn't provide enough justification for studying this phenomenon. And the hypothesis made by the author could not be tested properly. Thus the authors should mention exact methodology that has been followed with support from the literature.

Response 4: Chapter 3 (Research Methods) has been entirely re-edited. It has received a new structure within which we have better described and justified the purpose of our methodological approach. We adopted a framework based on triangulation of methods (1) The literature review, (2) Semi-structured interviews and expert workshops and (3) the focus group discussion in pilot towns. In this chapter, we also explain the background for 5 assumptions (which we previously imprecisely called hypotheses), as well as indicate and describe methods for their verification (Focus Group discussion).   

Point 5. No detailed discussion in the comparison among reviewed studies and showing the differences and incremental academic contributions between current study and them. This section lack discussion in the connections between existing studies and the current one. Therefore, it is not clear the academic contributions of the current study to the literature. Prove the results and find a linkage among results, research questions and hypothesis by using previous studies/literature review, whether your hypothesis accepted or rejected, matches with similar results and dissimilar. If dissimilar, mention what cause.

Response 5: We have significantly changed the structure of the "results" section, dividing it into subsections corresponding to the results achieved in the subsequent research steps. We hope that in this way the results have been made clearer, and the proposed set of indicators more clearly shows the connections with those that emerged from the analyzed studies and documents.

Point  6. The conclusion must shows the answer from the research objective and the policy implications and as a part of this section you can be concluded theoretical implication, future implication, and recommendation for future practice briefly. Need to include limitations of the study in this section.

Response 6 The conclusions have been fully rewritten and now better address the demands of the comment.

Reviewer 3 Report

As a first observation, I suggest to the authors to modify the abstract in order to convey with the rules needed to be followed when an abstract is written. Please do not mention the name of the project in the abstract - there is a special section in the paper where this information should be mentioned. Please add information related to the need of such a study, methods used and give a hint to the reader related to the results - if possible briefly present some valuable results or some indicators.

Figure 1 cannot be read - please provide a better figure in which the text is not blurry. If possible please add the expected results of each stage that are inputs to the next stage.

The selected papers in Table 2 can be moved in an annex as no additional information is provided in the paper either than the name of the paper and their associated code. Related to the 22 selected papers, please add in the paper details related on how the papers have been selected (e.g. keywords, exclusion criteria, database used, etc). and provide their names in English as it is hard to understand - for some of them - to which extent they are connected to the topic of the paper.

The following phrase is vague: "The second set of indicators consists of 20 items: each of the five dimensions linking built cultural heritage with quality of life was assigned four indicators. In order to create this set, the authors drew on the findings of the focus group meetings as well as the literature review." - please provide the list of the indicators.

Other from the description of the 10 towns in rows 209-222, I could not found any connection between the indicators and the 10 locations - can you please better state the connection and provide some quantitative and qualitative results to better highlight the results of the study?

Please add some limitations related to the selected indicators and their applicability in practice in similar studies.

Author Response

Thank you very much for your in-depth and inspiring comments. The manuscript has been thoroughly revised and we hope it will now better reflect and explain the idea of the research. Below we provide detailed responses to the comments.  

Point 1: As a first observation, I suggest to the authors to modify the abstract in order to convey with the rules needed to be followed when an abstract is written. Please do not mention the name of the project in the abstract - there is a special section in the paper where this information should be mentioned. Please add information related to the need of such a study, methods used and give a hint to the reader related to the results - if possible briefly present some valuable results or some indicators.

Response 1: The abstract has been entirely rewritten and restructured according to the suggestion. The name of the project in the abstract has been removed.  We hope that it now better reflects the theoretical and practical aspect of the study undertaken and also highlights the objective of the study. The abstract includes information about the research methods used, which is further developed in Chapter 3: Research Methods. It has been entirely re-edited and given a clearer structure.     

Point Figure 1 cannot be read - please provide a better figure in which the text is not blurry. If possible please add the expected results of each stage that are inputs to the next stage.

Response 2: We have resigned from Figure 1, although it was not due to its technical inadequacy. We provided the Editors with all the figures in appropriate format, required by the editorial board (both, in the text and in separate files). We do not know why the system transformed them in the way, they are not clear. Perhaps the separate files were not sent to you? In our document, the figures are sharp and visible.   

Point 3 The selected papers in Table 2 can be moved in an annex as no additional information is provided in the paper either than the name of the paper and their associated code.

Response 3: Table 2 has been moved to the Annex, according to the suggestion. 

Point  4 Related to the 22 selected papers, please add in the paper details related on how the papers have been selected (e.g. keywords, exclusion criteria, database used, etc). and provide their names in English as it is hard to understand - for some of them - to which extent they are connected to the topic of the paper.

Response 4: We have significantly changed the structure of the "research methods" part. It has received a new structure within which we have better described and justified the purpose of our methodological approach. In the section “Literature review” we have added the explanation on how the 22 papers for detailed analysis have been selected. We provide their full list with names in English in the Appendix.

Point 5: The following phrase is vague: "The second set of indicators consists of 20 items: each of the five dimensions linking built cultural heritage with quality of life was assigned four indicators. In order to create this set, the authors drew on the findings of the focus group meetings as well as the literature review." - please provide the list of the indicators.

Response 5: Since we have significantly changed the structure of the "results" section, the mentioned sentence has been removed. The list of indicators is provided in the Table 3   

Point 6: Other from the description of the 10 towns in rows 209-222, I could not found any connection between the indicators and the 10 locations - can you please better state the connection and provide some quantitative and qualitative results to better highlight the results of the study?

Response 6:  Our goal was not to examine the 10 pilot towns through the lens of the elaborated indicators. The focus group discussions in these towns served to test our earlier assumption that the impacts of cultural built heritage can be observed (and measured) in the five identified spheres (urban identity/sense of place; society; urban fabric, structure and spaces; services and facilities; economy). This checking step provided the basis for proposing a final list of indicators. Therefore, we cannot provide the results of the study for these towns, as their participation had a different purpose. At the same time, we thank you for the interesting suggestion to conduct a full survey in these cities in the future using the generated indicators.

Point 7 Please add some limitations related to the selected indicators and their applicability in practice in similar studies.

Response 7: The conclusions have been fully rewritten and we have supplemented them with a description of the limitations of the proposed set of indicators.  

Round 2

Reviewer 2 Report

I read with great interest the article “Using indicators to evaluate cultural heritage and the quality of life in small and medium-sized towns. The study on 10 towns 3 from Polish-German borderland”, which aims to investigate cultural heritage and the quality of life in small and medium-sized town from Polish-German borderland. Despite the interest of the topic, I do not believe that the article succeeds in delivering real insights from both a theoretical and a practical point of view. Results do not really build on statistical values (absent), literature, and the theoretical framework. 

Author Response

Thank you very much for your review. Your comments have been extremely valuable to us and we have tried, to the extent possible, to make appropriate corrections. Despite your concerns, we hope that if a decision is made to publish, our article will be of interest to Sustainability readers.

Sincerely,

Authors    

Reviewer 3 Report

I thank the authors for the provided response letter and the changes made to the manuscript. Even though the paper now reads better, the elements presented in the paper are not supported by the research conducted or by the review of the literature. There are still parts of the paper, such as section 3 where few information is provided related to the semi-structured interviews and the obtaining the experts opinion. The information related to how the experts have been selected and which are the elements that recommends them as experts in the field should be better highlighted. Also, in the revised version of the paper, section 4 is more like a long story and does not bring additional insight based on the used methodology.

Author Response

(The authors gave the same response as above.)
